# Interventions to Improve Medication Adherence in Patients with Schizophrenia or Bipolar Disorders: A Systematic Review and Meta-Analysis

**DOI:** 10.3390/ijerph181910213

**Published:** 2021-09-28

**Authors:** Elke Loots, Eva Goossens, Toke Vanwesemael, Manuel Morrens, Bart Van Rompaey, Tinne Dilles

**Affiliations:** 1Centre For Research and Innovation in Care (CRIC), Department of Nursing Science and Midwifery, Nurse and Pharmaceutical Care (NuPhaC), Faculty of Medicine and Health Sciences, University of Antwerp, 2610 Antwerp, Belgium; eva.goossens@uantwerpen.be (E.G.); toke.vanwesemael@uantwerpen.be (T.V.); Bart.VanRompaey@uantwerpen.be (B.V.R.); Tinne.Dilles@uantwerpen.be (T.D.); 2Department of Public Health and Primary Care, University of Leuven, 3000 Leuven, Belgium; 3Research Foundation Flanders (FWO), 1000 Brussels, Belgium; 4Department of Patient Care, Antwerp University Hospital (UZA), 2610 Antwerp, Belgium; 5Faculty of Medicine and Health Sciences, Collaborative Antwerp Psychiatric Research Institute (CAPRI), University of Antwerp, 2610 Antwerp, Belgium; manuel.morrens@uantwerpen.be

**Keywords:** adherence, interventions, schizophrenia, bipolar disorders, compliance

## Abstract

Adherence to prescribed medication regimes improves outcomes for patients with severe mental illness such as schizophrenia or bipolar disorders. The aim of this systematic review and meta-analysis was to compare the effectiveness among interventions to improve medication adherence in patients with schizophrenia or bipolar disorders. Literature published in the last decade was searched for interventions studies to improve adherence in patients with schizophrenia or a bipolar disorder. Interventions were categorised on the basis of type, and the context and effectiveness of the interventions were described. Two review authors independently extracted and assessed data, following criteria outlined by the Cochrane Handbook for Systematic Reviews of Interventions. The GRADEPro (McMaster University, 2020, Ontario, Canada) was used for assessing the quality of the evidence. Twenty-three publications met the selection criteria. Different types of interventions aiming to improve adherence were tested: educational, behavioural, family-based, technological, or a combination of previous types. Meta-analysis could be performed for 10 interventions. When considered separately by subgroups on the basis of intervention type, no significant differences were found in adherence among interventions (*p* = 0.29; I^2^ = 19.9%). This review concluded that successful interventions used a combination of behavioural and educational approaches that seem easy to implement in daily practice.

## 1. Introduction

Psychiatric disorders are a public health challenge and comprise 13% of the total global disease burden [1]. Schizophrenia and bipolar disorders are severe major psychiatric disorders, with schizophrenia affecting about 23 million people and bipolar disorders affecting about 60 million people worldwide [2]. Together with psycho-education, pharmacotherapy is often the first line of treatment of these major psychiatric disorders. Hence, maintaining medication adherence is crucial [3,4,5,6]. Varieties of risk factors for disease relapse have been reported, including medication non-adherence, substance abuse and stressful life events. A recent systematic review analysed risk factors for relapse in the early course of psychosis in patients with schizophrenia [7]. Among all associated factors, non-adherence appeared to be the strongest predictor for relapse. Discontinuing antipsychotic pharmacotherapy increased the risk of relapse by almost five times [8].

Non-adherence is highly prevalent, ranging between 63–74% in patients with schizophrenia and about 50% in patients with bipolar disorders [9,10,11]. About 25% of patients discontinue their medication within the first week after discharge from inpatient treatment [12]. Non-adherence puts patients at risk for exacerbations of psychosis and relapse resulting in hospital visits and admission [6,13,14,15,16,17,18,19,20,21,22]. Relapse rates appear to be high at 78–82% for schizophrenia and 60% for bipolar disorders [23,24]. Non-adherent patients have an average relapse risk that is 3.7 times greater than adherent patients [16].

Medication adherence is, however, a complex behaviour comprising a series of interrelated steps involving patients, their providers, and the healthcare system [3]. Adherence to medications can be defined as “*the process by which patients take their medication as prescribed, described by three quantifiable phases: initiation, implementation, and discontinuation*” [25]. Non-adherence is defined as taking less than 80% of prescribed doses. This cut-off has validity in predicting subsequent hospitalisation [26].

Patient-related factors impeding medication adherence in schizophrenia or bipolar disorders include medication side effects, lack of insight into the illness, cognitive dysfunction, regimen complexity and substance use [7,27,28,29].

A variety of interventions have been used to improve medication adherence, such as cognitive behavioural therapy, psychoeducation, family interventions, motivational interviewing techniques, and mixed interventions [30,31,32,33,34,35]. To date, however, a detailed overview of the effectiveness of these interventions at improving medication adherence in patients affected by schizophrenia or bipolar disorders is lacking.

Hence, the aim of this systematic review and meta-analysis was to explore the impact of interventions on medication adherence in patients with schizophrenia or bipolar disorders in patients with schizophrenia or bipolar disorders.

## 2. Methods

### 2.1. Overview

A systematic review, comprising a meta-analysis, was performed including a detailed assessment of the quality of evidence. Furthermore, the certainty of evidence related to interventions, designed to improve medication adherence in patients with schizophrenia or bipolar disorders, was systematically rated using the GRADE approach [36]. The review protocol was registered at PROSPERO (PROSPERO 2020 CRD42020153237).

### 2.2. Search Methods for Identification of Studies

#### Electronic Searches

The review focused on studies examining the effectiveness of interventions aimed at improving adherence in patients with schizophrenia or bipolar disorders. PubMed and Web of Science were systematically reviewed for relevant intervention studies published between 2009 and 2019. Studies had to be published in Dutch, English or French. Details on the applied search string can be found in Table 1. Using the snowball method, reference lists of all retrieved articles were screened to identify additional publications.

### 2.3. Selection Criteria

#### Types of Studies and Study Population

Full-text (quasi-)randomised controlled trials and prospective trials, comparing adherence-enhancing interventions versus no or other interventions, were selected. Control groups or treatment as usual (TAU) should have received no intervention, other interventions, or usual care. The study population consisted of (i) adults (≥18 years); (ii) diagnosed with schizophrenia, schizoaffective disorder, or Bipolar I/II disorder, according to an official classification system such as the Diagnostic and Statistical Manual of Mental Disorders (DSM-criteria) or International Classification of Diseases (ICD) and had to be made by a physician; and (iii) cared for within in- or outpatient setting(s) [37,38]. Studies that examined patients with a first episode of psychosis, or patients with neurological comorbidities, such as mental retardation, were excluded. All retrieved hits were initially screened for eligibility on the basis of title and abstract by two independent researchers (EL, TVW). Subsequently, a full text appraisal was performed. Two authors (EL, EG) independently decided on inclusion or exclusion of selected studies. All discrepancies were discussed until consensus was achieved. Detailed information about the search strategies can be found in Figure 1.

### 2.4. Outcome Measures

The outcome was medication adherence, irrespective of the definition of adherence used in the manuscripts. All studies investigating adherence as an outcome were included. No distinction was made among studies investigating adherence as either a primary or secondary outcome. Studies could employ both objective metrics of adherence, such as pharmacy claims, pill counts or blood plasma concentration levels, as well as subjective measures such as clinician-rated or self-reported measures of medication adherence using standardised and validated assessments. The effects of the different interventions were assessed using effect sizes (Cohen’s d). In line with Cohen’s classification, effect sizes were divided into five levels: trivial (Cohen’s d ≤ 0.2), small (Cohen’s d > 0.2), moderate (Cohen’s d > 0.5), large (Cohen’s d > 0.8) and very large (Cohen’s d > 1.3) [39,40].

### 2.5. Data Extraction and Management

Two authors (EL, EG) extracted data until the end of November 2019, including details of study methodology, outcome measurement(s), demographics and clinical sample characteristics, eligibility criteria, details of the intervention, baseline and post-intervention results, methods of analysis and follow-up time. Information was recorded in the ‘Cochrane Airways’ and authors were contacted in case of missing information or when clarification was needed [41].

### 2.6. Risk of Bias Assessment

Two authors (EL, EG) independently assessed the methodological quality of selected studies using the Cochrane Risk of bias tool version 1.0, described in the Cochrane Handbook for Systematic Reviews of Interventions. For each respective domain, the risk of bias was assessed as either high, low or unclear. Furthermore, the studies’ overall risk of bias was determined on the basis of the following criteria as either low [i.e., low risk of bias for all domains), unclear (i.e., unclear risk of bias for one or more domains) or high (i.e., high risk of bias for one or more domains) [42].

In addition, the overall strength of evidence on outcomes was evaluated using the GRADE approach [36]. The outcomes included effects on adherence on the basis of behavioural, educational and mixed interventions. The GRADE approach considers evidence from randomised controlled trials as high quality, although this level may be downgraded on the basis of five areas of consideration: design, consistency across studies, directness of the evidence, precision of estimates and presence of publication bias [42].

### 2.7. Data Synthesis

Firstly, the clinical heterogeneousness of studies was determined on the basis of their clinical characteristics including the intervention, control group, outcome assessment and follow-up window. When similarity among studies allowed data pooling, the Review Manager 5.3 data analysis tool was used for the assessment of statistical heterogeneity, as indicated in the forest plots measuring the treatment effect. I^2^ and Chi^2^ statistics were applied to determine statistical heterogeneity. Data were considered heterogeneous when *p*-value was ≤0.10. I^2^ thresholds, as described in the Cochrane Handbook, were used as a guide for interpretation. Furthermore, we use the I^2^ statistic to quantify the amount of heterogeneity. We considered an I^2^ < 40% as low heterogeneity; 0% to 40%: might not be important, 30% to 60%: may represent moderate heterogeneity, 50% to 90%: may represent substantial heterogeneity, 75% to 100%: considerable heterogeneity [42].

Results in terms of adherence concerning intervention compared to treatment as usual (TAU) were used. Forest plots were used to present results obtained from the meta-analysis. Narrative syntheses were used when studies were not eligible for meta-analysis. These data are presented in Appendix A.

## 3. Results

### 3.1. Study Characteristics

#### 3.1.1. Results of the Search

The systematic search yielded 2584 results. Of those, 1568 studies were retrieved from Web of Science and 1016 from PubMed. After removal of 114 duplicates, 2470 references were screened on the basis of title and abstract. Sixty-five studies were assessed on the basis of full text, of which 42 were excluded. Reasons for exclusion were: full text was unavailable (*n* = 7), studies did not contain any data on adherence (*n* = 23), including other study populations (*n* = 2), no interventional study design (*n* = 8) and segmented publications (*n* = 2). Twenty-three studies were included in this systematic review and meta-analysis. A selection flow chart is provided at Figure 1.

All included studies were randomised controlled trials and compared intervention versus no intervention or another intervention, except for one study that compared an educational intervention, a behavioural intervention and a control group, respectively [43]. The follow-up time ranged from one month to 30 months (see Appendix A).

#### 3.1.2. Participants and Setting

A total of 4238 participants, ranging from 30 to 1268 per study, were included. Of the total sample, 2967 patients (70%) were patients diagnosed with schizophrenia or schizoaffective disorders and 1271 patients (30%) were diagnosed with a bipolar disorder. Studies were performed across three continents: eight studies in Asia [44,45,46,47,48,49,50,51], ten studies in Europe [21,52,53,54,55,56,57,58,59,60] and six in North America [19,27,43,61,62,63]. Study settings were categorised on the basis of the setting where interventions were initiated as part of the patient’s healthcare journey. Most of the interventions were conducted at outpatient community mental health centres (65%) or in psychiatric hospitals (35%).

A range of complex interventions was used across selected studies including the provision of patient education and information, family involvement, intensified patient care (e.g., sending out reminders, telephone calls), complex behavioural approaches (e.g., increasing motivation by interviews, group sessions) and mixed therapies (Table 2). Due to the heterogeneous nature of the interventions, three categories were used including behavioural, educational or mixed (i.e., behavioural and educational approach) interventions. Nine studies examined 11 behavioural interventions, 11 studies involved educational interventions focussing on medication and treatment, and six studies combined educational and behavioural elements.

A range of behavioural interventions were used: six interventions focused on pharmacotherapy combined with text messages or telephone calls [19,49,59,61], three interventions practised motivational interviewing [21,45,53], one study used cognitive behavioural therapy [57] and one study provided participants with electronic reminders [63]. Education sessions were organised in groups or one-on-one with a nurse or another healthcare provider [43,44,48,52,53,54,55,56,58,62,64]. Participants received information concerning medication strategies such as the use of a pill container, medication, symptoms and had the opportunity to have a ‘Question and Answer’ (Q&A) session with their healthcare provider. Five interventions combined education and motivational interviewing related to medication use [43,46,47,50,60]. One intervention combined medication skills training, family involvement and cognitive behavioural therapy [64].

### 3.2. Medication Adherence Assessment

Three categories of adherence assessment were identified, including (i) direct measures, such as blood serum levels, (ii) indirect measures such as pill counts, electronic monitoring, prescription refill rate, and (iii) subjective measures such as patients’ and nurses’ self-report adherence rating scales or interviews. Three studies used direct measures such as blood serum levels [50,54,60]. Indirect measures included use of pill counts [60,61,63] and an electronic monitoring cap recording the number and timing of bottle openings [63,64].

Subjective measures such as the Compliance Rating Scale [44,52], the Medication Adherence Questionnaire [53,59], the Medication Adherence Rating Scale [19,48,50,56], the Morisky scale [21,49,51,55,58], the Visual Analog Scale for Assessing Treatment Compliance [46], the Stephenson Medical Adherence Questionnaire [57], the composite adherence measure and the medication possession ratios were used [62]. Two studies used an unknown Likert scale assessment tool [43,45] and two studies were unclear about the assessment tool used [47,65].

Adherence rates were reported as mean or median scores or percentages or percentages of complete doses taken or assessment tool scores. Follow-up time ranged from one to 84 months. Most of the studies defined adherence as taking more than 70% of prescribed doses. Six studies did not provide any definition for adherence [19,50,56,60,61,64].

### 3.3. Effectiveness of Interventions

#### 3.3.1. Behavioural Interventions

Six of nine included studies compared a behavioural intervention to usual care [19,21,45,49,51,57,59] and three studies compared a behavioural intervention versus other interventions [53,61,63]. In all studies, the outcome was adherence. All interventions aimed at improving medication adherence; however, the intervention was unclear [21], one study focused on general health [53], and one on diagnosis and identification of recovery-informed therapy goals [57]. Details on the main findings, related to the effect of behavioural interventions on adherence, can be found in Table 3.

SMS interventions were associated with significant improvements in medication adherence after three-month follow-up with a moderate effect size of 0.64 (*p* < 0.001) and after six-month follow-up (*p* = 0.04) [49,59].

Motivational interviewing was performed in two studies. One study recruited 114 patients with schizophrenia with poor adherence to medication. The intervention was based on motivational interviewing in eight sessions during a four-month program. Medication adherence in the intervention group showed a significantly greater improvement at 6-month follow-up, with a moderate effect size of 0.72, as compared to TAU (*p* = 0.007) [45].

The PharmCAT individualised intervention used signs, alarms, pill containers and checklists to improve medication adherence. Participants were seen once weekly at home. The Med-eMonitor intervention consisted of a therapist who programmed prescription information into the device, and set the device up at home to fit into the patient’s routine (e.g., set alarm to take medication). These two behavioural interventions showed a statistically significant enhancement in medication adherence at all time points during treatment and after nine-month follow-up as compared to TAU (*p* < 0.001). The PharmCAT reached a very large effect size of 1.03 and the Med-eMonitor a large effect size of 0.98. Differences between the two behavioural interventions were not significant (*p* > 0.43) [63].

In summary, 6 out of 12 behavioural interventions showed a statistically significant improvement on adherence. These interventions used an individualised approach to enhancing medication adherence. Motivational interviewing, daily SMS reminders, medication reminders at patients’ homes and medication self-management training were beneficial for patients’ adherence [21,45,49,59,63]. SMS and phone calls focused on problem solving strategies and cognitive behavioural therapy did not prove beneficial for patients’ adherence.

#### 3.3.2. Education

Nine of the 11 included studies compared an educational intervention to usual care [43,44,48,52,54,55,56,58,62] and three compared it to other interventions [27,43,53]. Eight studies investigated the effect of an intervention focusing on knowledge about medication and symptoms. Two studies were unclear about the content of the intervention [52,56] and one study focused on education covering the topic of general health [56].

Eight of eleven educational interventions had a statistically significant improvement of adherence [43,44,48,55,56,62,64]. Education sessions focused on diagnosis, symptoms, medication, relapse, Q&A, medication skills and medication adherence. These educational interventions were individualised and were provided on a one-on-one basis with a healthcare provider or in small group sessions. Education focused on stress reduction and problem-solving strategies did not show beneficial effects on patients’ adherence. Details on the main findings related to the effect of educational interventions on adherence can be found in Table 4.

#### 3.3.3. Mixed Interventions

Four of six included studies compared mixed interventions to usual care [46,47,50,60] and two studies compared it to other interventions [30,43,64]. Four studies focused their mixed intervention on medication [43,46,60,64] and two studies did not provide sufficient detail about the content of the intervention [47,50]. Details on the main findings can be found in Table 5.

One mixed intervention combined education of patients and family members with motivational interviewing. Using the Medication Adherence Rating Scale, the intervention group showed a significantly higher medication adherence compared to TAU, both at one (*p* < 0.001) and six months (*p* < 0.001) post-intervention (large effect size of 0.84). Analysis of the objective measures of medication adherence, such as plasma level of mood stabilisers indicated that participants in TAU had slightly decreased levels at six months post-intervention, suggesting they may not have been adhering to their medication regimen. In contrast, the intervention group had increased levels at six-month follow-up supporting the beneficial effects of the intervention suggested by self-report measure of adherence. After controlling for study centre and repeated measurements, the intervention group had significantly higher plasma levels of mood stabilisers as TAU at one (*p* < 0.001) and six months (*p* < 0.001) post- intervention [50].

In total, five of six mixed interventions had a positive impact on adherence. These mixed interventions were focused on an individualised approach of medication adherence. Interventions involving patients’ family members, medication preparing in a controlled environment and individualised interventions with medication techniques and an adequate follow-up with telephone calls were beneficial for patients’ adherence. There was not a beneficial effect on adherence from the combination of motivational interviewing and cognitive behavioural therapy [43,46].

### 3.4. Effects on Adherence

Four interventions of eleven studies reported effect sizes. Additionally, 11 interventions reported sufficient information to calculate effect sizes. For these 15 interventions, effect sizes could be appreciated as very large for one intervention [50], large for three interventions [50,63,64], moderate for six [45,46,49,52,55,62], small for three [19,53,61] and only a trivial effect for two interventions [53,61]. Fourteen interventions did not report sufficient information to calculate effect sizes.

Meta-analysis could be performed for 10 interventions in eight studies that involved dichotomous measures (Figure 2). The analysis was divided into three categories on the basis of the type of intervention provided: behavioural interventions (*n* = 1 study with two different behavioural interventions), educational interventions (*n* = 5 studies) or mixed interventions (*n* = 3 studies). The respective forest plots (presented on a logarithmic scale) showed pooled treatment effects of interventions in all categories as compared with usual care (TAU) for adherence at short-term and long-term follow-up (i.e., one month until 84 months). When considered separately by subgroups on the basis of intervention type, no significant differences were found in adherence between interventions (*p* = 0.29; I^2^ = 19.9%).

A significant difference in adherence rates was found between behavioural interventions and TAU; 92% versus 72% adherence in the PharmCAT intervention and 89% versus 72% in the Med-eMonitor intervention. Meta-analysis using a random-effects model estimated an odds ratio of 3.65 (95%CI: 1.60 to 8.31).

Five studies were included in meta-analysis for educational interventions. There was considerable heterogeneity (I^2^ = 72%). Pooling of data used dichotomous measures of adherence at 2.5 to 84-month follow-up range involving 408 participants. Using a random-effects model, pooled results showed that adherence was greater in the intervention group (estimated odds ratio = 4.86; 95%CI: 2.96 to 7.97). The educational intervention of Bäuml (2016) [54] had no significant improvement on adherence when comparing the intervention group with TAU at 84-month follow-up (95%CI: 0.19 to 5.99).

Regarding the effect of mixed interventions, data of 1451 participants were pooled using dichotomous measures of adherence at 1- to 24-month follow-up. Using a random effects model, meta-analysis showed mixed interventions increased the proportion of adherent patients (estimated odds ratio= 2.27; 95%CI: 1.44 to 3.59). There was no evidence of significant heterogeneity (I^2^ = 0%).

### 3.5. Risk of Bias

The risk of bias of each included study is summarised in Figure 3 and Figure 4. Descriptions for each respective domain are provided below.

#### 3.5.1. Allocation

Risk of bias for random sequence generation was low in 16 studies (70%), unclear in five studies (22%) and high in two studies (8%). Eight trials used computer-generated randomisation, which we considered to be an adequate randomisation procedure [43,44,45,49,50,53,60,63].

#### 3.5.2. Blinding

Six studies (26%) were considered to have low risk of performance bias, 12 studies (52%) were unclear about blinding of participants and personnel, and five studies (22%) were considered to have high risk of performance bias. Blinding of healthcare providers was reported in six studies [45,47,48,52,53,57]. None of the studies reported blinding of participants to the intervention they were receiving, as this was not deemed feasible given the nature of the interventions. Eight studies reported blinding of outcome assessors and hence were considered to have a low risk of detection bias [43,45,47,48,49,51,53,57,60].

#### 3.5.3. Incomplete Outcome Data

Twelve studies (52%) were assessed as having low risk of bias mainly due to low attrition rates and the use of intention-to-treat analysis (ITT). Attrition >20% was considered to indicate a high risk of bias. Nine studies (39%) were considered to have incomplete outcome data because of high attrition rates, and therefore identified as having a high risk of attrition bias. Two studies (9%) did not report information on missing data [19,64].

#### 3.5.4. Selective Reporting

Selective outcome reporting bias occurred if adherence frequency was measured and analysed but was not reported in the study results. One study (4%) was considered to have a high risk of reporting bias due to risk of multiple testing [21]. Six studies (25%) reported their results insufficiently [19,43,44,47,52,56,61].

Seventeen studies (71%) were assessed as having a low risk of selective reporting bias due to transparency in results and publishing of all expected outcomes.

#### 3.5.5. Other Potential Sources of Bias

Other potential sources of bias contained limited follow-up, self-reported assessment tools, small sample sizes and an unclear assessment tool for adherence. Risk of bias for other potential sources of bias was low in 13 studies (54%) and high in 10 studies (42%). Two studies (4%] were found to be free of other sources of bias [50, 64). Six studies reported the combination of a limited follow-up time and a self-reported assessment tool [19,44,46,52,56,58]. Two studies performed appropriate sample size calculations in combination with limited follow-up [57,61,65]. Seven studies only reported a self-reported assessment tool [21,48,49,51,53,57,59] and two studies contained a limited follow-up [60,63,65]. Four studies reported insufficient information about their assessment tool [43,45,47,62].

#### 3.5.6. Overall Strength of Evidence (GRADE)

The studies were, overall, low in quality (see Table 6); some studies appeared to have a considerable risk of bias. Additionally, the length of follow-up applied in the respective studies ranged from one to 84 months. Short-term follow-up makes it difficult to ascertain whether interventions with promising adherence-improving effects can safeguard and maintain their effects over time. The nature of the studied interventions implied that blinding of participants and personnel was not possible. Hence, we did not downgrade the evidence for lack of blinding.

## 4. Discussion

This is the first systematic review providing a synthesis of the effectiveness of interventions improving medication adherence in patients with schizophrenia or bipolar disorders, including a meta-analysis.

### 4.1. Summary of Main Results

On the basis of a synthesis of 23 studies, a total of 28 different, complex and heterogeneous interventions were identified. These interventions comprised behavioural, educational and mixed interventions, and were compared versus usual care or other types of interventions. Various interventions produced favourable results regardless of type, duration or setting. On the basis of this detailed assessment, motivational interviewing, daily SMS medication reminders, medication reminders at patients’ home, education sessions focused on diagnosis, symptoms, medication and relapse were found to be beneficial for patients’ adherence [49,59,63]. Educational interventions were individualised and were provided on a one-on-one basis with a healthcare provider or in small group sessions [43,44,48,62,64]. The interventions with the strongest of body of evidence were two interventions combining motivational interviewing techniques with patient-tailored education [50,60]. These two studies had a very low risk of bias and used a combination of two or more adherence measurement tools, including serum levels. One of the mixed interventions found to be effective had a large effect size at six-month follow-up and combined education of patients and family members with motivational interviewing [50]. Family members and patients were given information about symptoms, prognosis of the condition, as well as the prescribed medication and their possible side effects.

Each family member was provided information about the importance of medication adherence and the risks of discontinuing these medications. At the end of the sessions, family members were given a booklet with information about the diagnosis and possible treatments. Unfortunately, interventions aiming to include and target interventional components to family members are challenging to implement in everyday practice and generally create a high workload. One intervention used an intensive training program comprising one-to-one lessons provided by skilled nurses. Participants should learn to prepare their medication themselves during the hospital stay in the same way they are expected to do it autonomously after discharge [60]. Unfortunately, this intervention was only tested at short-term follow-up of one month. Our review concluded the difficulty of evaluating of the effectiveness of all interventions against each other due to the heterogeneous and complex nature of the interventions and variations in adherence measures (i.e., different follow-up range, and various pathologies). Our results showed the use of short duration interventions produces equally favorable results as long-term interventions. Problems with adherence are recurrent, and therefore booster sessions are needed to maintain adherence.

### 4.2. Long-Term Follow-Up

Studies including adequate and extensive follow-up periods are important, as researchers need to measure the immediate effects of their intervention(s) on adherence, but also intermediate and long-term effects. Education focusing on medication, symptoms, treatment and diagnosis resulted in achieving favourable results on adherence at six-month follow-up with a large effect size [64] and 12-month follow-up with moderate effect sizes [48,62]. A 12-month intervention focused on medication adherence, including education and motivational interviewing, resulting in favourable results on adherence at 12-month follow-up with a large effect size, but not at 24-month follow-up. Repeating the intervention may improve this result [43]. One behavioural intervention study provided a long-term follow-up of nine months with a large effect size. This intervention used signs, alarms, pill containers and checklists to improve medication adherence. Participants were seen once weekly at home [63]. Two other studies, where motivational interviewing focused on medication and medication changes were used, achieved favourable results on adherence at six-month follow-up with a moderate effect size [21,45].

### 4.3. Assessment of Adherence

No single measurement method can be regarded as the best available approach given the various patient-related factors (i.e., lack of disease insight, and forgetfulness). Hence, the use of multiple measurement methods of adherence is highly recommended. The wide variety of settings, intervention types, medications prescribed, adherence measures and follow-up time precluded summarising findings to reach reliable general conclusions.

### 4.4. Critical Appraisal of the Methodology

The strength of our review is the performance of a thorough literature search, which was performed using a strict and systematic approach when selecting studies for inclusion, as well as extracting and analysing data. Furthermore, the body of evidence was evaluated using the GRADE approach for the outcome of medication adherence (see Table 6). Twelve authors were contacted to clarify missing information concerning the interventions and data results. Unfortunately, we received the missing information from only two authors [19,63]. The studies were overall low in quality (see Table 6); some studies appeared to have a considerable risk of bias. Additionally, the length of follow-up applied in the respective studies ranged from one to 84 months.

A well-known problem in the literature is the lack of uniformity in the terminology used to describe deviation from prescribed medication regimens. The conceptual definitions vary resulting in conceptual confusion, which adds to the methodological weakness in this field [25]. This heterogeneity of operational definitions for medication (non-)adherence was the main obstacle experienced when comparing study findings in this systematic review and meta-analysis. The included interventions differed not only in terms of interventional components, but also in terms of their comparison group (no intervention or other intervention), duration of interventions and follow-up time. The performance of a meta-analysis was only possible for 10 interventions described in eight studies.

Concerns could be raised related to inconsistencies due to the heterogeneous and complex nature of the interventions and variations in outcome measures (i.e., follow-up range and methods of measuring adherence). Sixteen out of 24 studies followed patients up for six months or more. Most studies used patient self-reported measures, which are known to overestimate adherence rates [66,67]. With regard to the problem of non-adherence, the different rates reported in the publications may partly reflect methodological obstacles concerning the difficulty to relabelling measurements reported in the respective papers. A reliable measurement is a prerequisite for addressing non-adherence. Definitely, no such method exists at this moment. Direct measurements such as blood or urine drug levels are less subjective to bias as compared to indirect measurements such as self-reports, pill counts or refill rates. Practically every method aiming to determine adherence rates has specific limitations [63,68].

Although interventions were categorised as either having a behavioural, educational or mixed interventional focus, low to high heterogeneity was evident contributing to the limited certainty of results derived from literature. Concerns related to imprecision were present for behavioural and educational interventions, for which participant numbers were low and confidence intervals were wide. In line with previously published literature, our systematic review revealed that currently high-quality evidence is lacking addressing the effectiveness of interventions improving medication adherence in patients with schizophrenia or bipolar disorders. Furthermore, variabilities in the study methodology applied, interventions used, and outcome measures selected made it difficult to draw any firm conclusions in terms of the most effective intervention improving medication adherence in patients with schizophrenia or bipolar disorders. However, it is difficult to establish the relationship between the different interventions and adherence, as different measurement points and definitions of adherence were used.

### 4.5. Future Prospects

Our findings emphasise the need for future studies using mixed interventions. These interventions comprising elements of education, motivational interviewing and medication self-management, evaluating adherence rates by using a combination of measurement tools during longer-term follow-up times. The use of checklists, pill containers, one-to-one medication education and medication self-management techniques are hypothesised to result in favourable outcomes. Researchers should minimise the risk of bias by using suitable randomisation techniques, allocation concealment and double blinding techniques.

Researchers should strongly consider prospective trial registration and publication of study protocols using standard reporting checklists such as the Standard Protocol Items: Recommendations for Interventional Trials [66]. This will help to ensure clearer and more consistent reporting of outcome variables impacting medication adherence. In terms of study design, studies of duration are important, as researchers need to be able to made valid assessments of the short-term, mid-term and long-term effects of their intervention on adherence.

## 5. Conclusions

Our review is the first to provide a synthesis on the effectiveness of interventions aiming to improve medication adherence in patients with schizophrenia or bipolar disorders. Successful interventions used a combination of educational and behavioural strategies. The combined use of education sessions focusing on diagnosis, symptoms, medication and relapse, with medication reminders at patients’ home and an intensive training program provided on a one-to-one basis by skilled nurses can improve medication adherence. Furthermore, such mixed interventions are deemed feasible to implement in daily practice. Our findings emphasise the need for future studies evaluating the effectiveness of such mixed interventions. These interventions comprising elements of education, motivational interviewing and medication self-management, evaluating adherence rates using a combination of measurement tools during longer-term follow-up periods.

## Figures and Tables

**Figure 1 ijerph-18-10213-f001:**
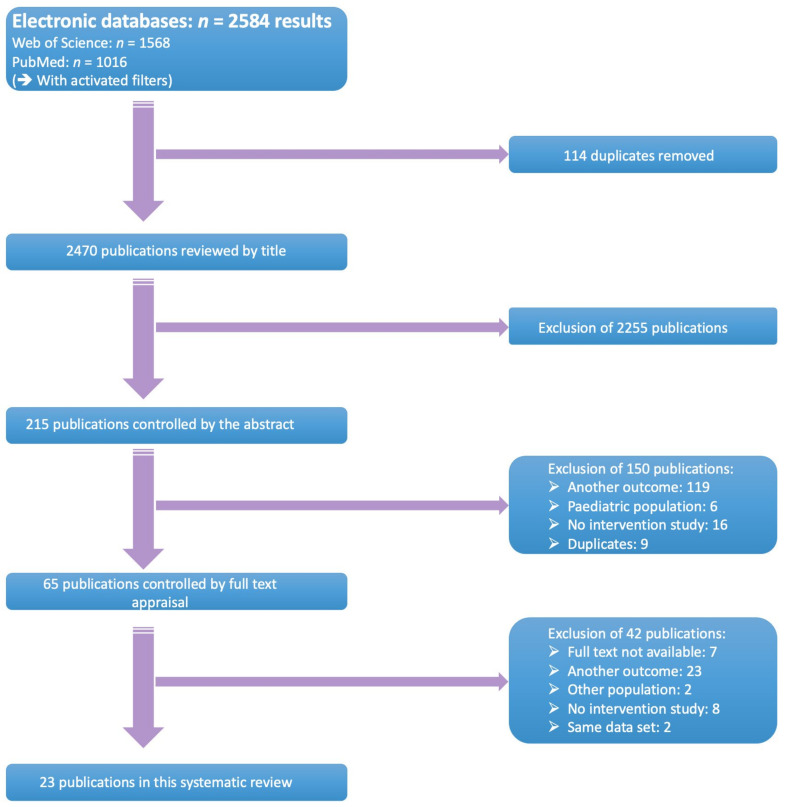
Selection flowchart.

**Figure 2 ijerph-18-10213-f002:**
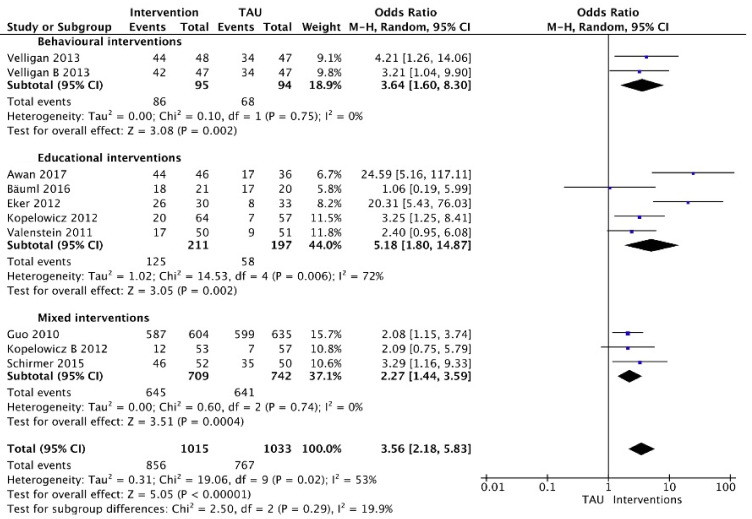
Interventions versus usual care grouped by type of intervention (dichotomous).

**Figure 3 ijerph-18-10213-f003:**
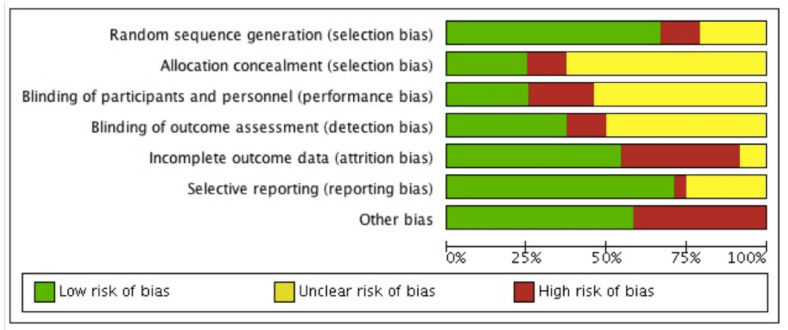
Risk of bias graph.

**Figure 4 ijerph-18-10213-f004:**
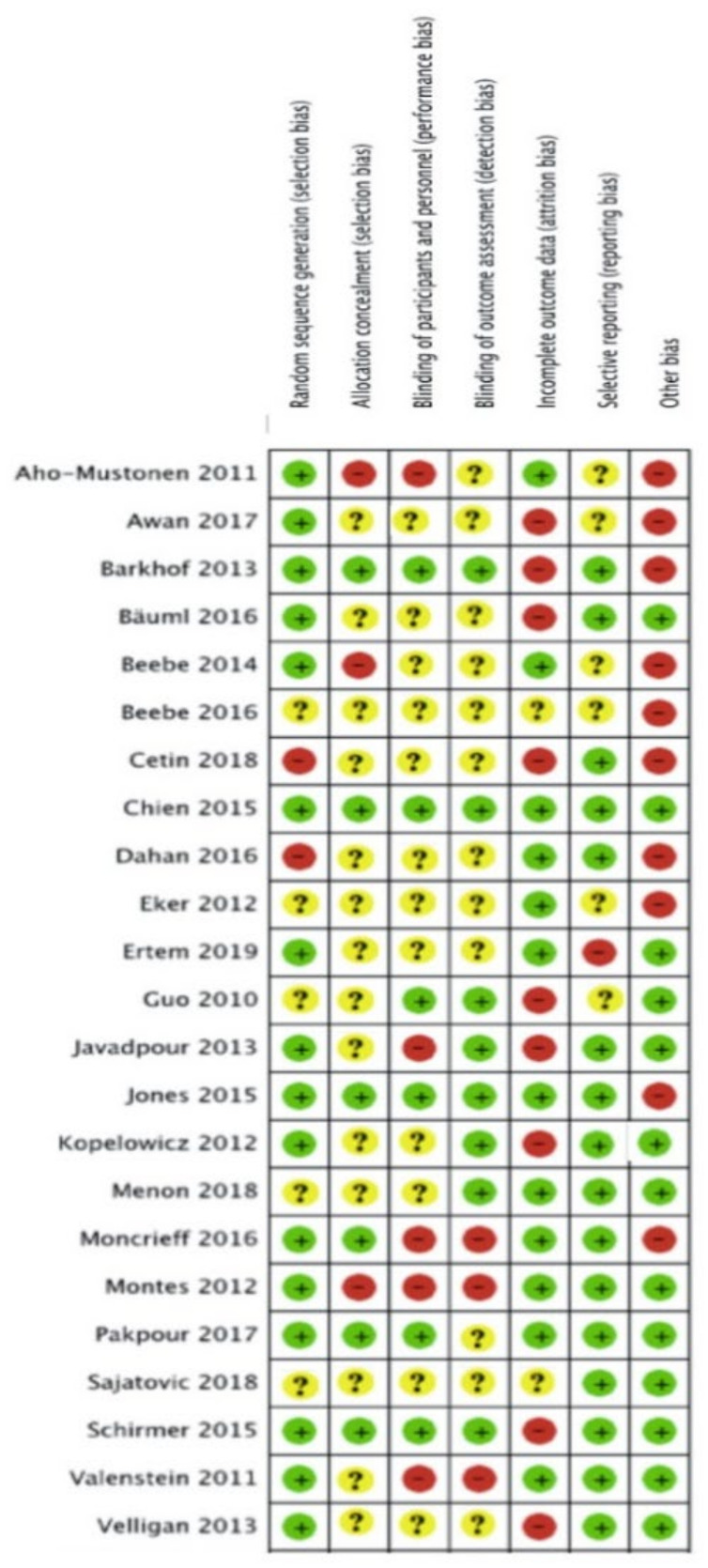
Risk of bias assessment of included studies using the Cochrane Risk of bias tool.

**Table 1 ijerph-18-10213-t001:** Search string.

Concept	Keywords ^a^	Keywords ^b^
1. Outcome: Medication adherence	“Medication Adherence”[Mesh]) OR medication adherence[Title/Abstract]) OR medication compliance[Title/Abstract]) OR medication persistence[Title/Abstract]) OR medication training[Title/Abstract]) OR medication management[Title/Abstract]) AND	TITLE: (medication adherence) OR TITLE: (medication compliance) OR TITLE: (medication persistence) OR TITLE: (medication training) OR TITLE: (medication management) AND
2. Participant: Patients with schizophrenia or bipolar disorders	“(schizophreni *) OR bipolar disorder *) OR bipolar mood disorder *) OR schizoaffective disorder *) OR “Schizophrenia”[Mesh]) OR “Bipolar Disorder”[Mesh]) AND	TITLE: (schizophren *) OR TITLE: (bipolar disorder *) AND
3. Exposure	intervention*[Title/Abstract]) NOT protocol[Title]	TOPIC: (intervention *) NOT TITLE: (protocol *)
4. Filters *	Clinical Study, Clinical Trial, Comparative Study, Controlled Clinical Trial, Pragmatic Clinical Trial, Randomized Controlled Trial	psychiatry, medicine general internal or nursing

^a^ Used in PubMed; ^b^ Used in Web of Science; * The filters were activated after entering the search terms.

**Table 2 ijerph-18-10213-t002:** Overview of the types of interventions included in selected literature.

Intervention Categories	Behavioural	Educational	Mixed
Examples	- Motivational interviewing - SMS ^1^ reminders - Alarms - Checklists - MEMS^® 2^ - Meetings - Family involvement	- Education sessions - Website tool	Combination behavioural and educational intervention(s)
Number of interventions	11	11	6

^1^ Short message service; ^2^ Medication Event Monitoring System.

**Table 3 ijerph-18-10213-t003:** Summary of results on the effectiveness of behavioural interventions.

Reference	Assessment Methods	Follow-Up	Number of Participants	Cohen’s d	Study Results
Barkhof (2013) [53]	Medication Adherence Questionnaire	Baseline, 6 and 12 months	Motivational interviewing (*n* = 55) Health education (*n* = 59)	0.29	No significant differences between motivational interviewing and health education on 6- and 12-month follow-up (*p* = 0.34).
Beebe (2014) [61]	Pill counts	Baseline and 3 months	Telephone call (*n* = 10) SMS (*n* = 10) Telephone + SMS (*n* = 10)	−0.19 0.36 −0.70	No significant difference in adherence was noted between the groups on the basis of pill counts (*p* = 0.31).
Beebe (2016) [19]	Medication Adherence Rating Scale	Baseline and 3 months	*n* = 140	0.29	Self-reported medication adherence was higher in the intervention group after 3 months but the differences were not statistically significant.
Chien Tong (2015) [45]	Unclear	Baseline, immediately post intervention and 6 months post intervention	Motivational interviewing (*n* = 57) TAU (*n* = 57)	0.72	The medication adherence of the motivational interviewing group showed a significantly greater improvement over time with a moderate effect size of 0.72, when compared with the control group (*p* = 0.007).
Ertem (2018) [21]	Morisky scale	Baseline, immediately post intervention, 3 and 6 months	Motivational interviewing (*n* = 20) TAU (*n* = 20)	*n*/A	Participants in the motivational interviewing group showed a significant improvement after 3-month follow-up post intervention (*p* < 0.001) and 6-month follow-up (*p* < 0.001).
Jones (2015) [57]	Stephenson Medical Adherence Questionnaire	Baseline, 6, 12 and 15 months post intervention	Cognitive behavioural therapy (*n* = 34) TAU (*n* = 33)	*n*/A	No significant difference in adherence was noted between the two groups on the basis of self-reports at baseline, 6- and 12-month follow-up.
Menon (2018) [49]	Morisky scale	3 months	SMS intervention (*n* = 62) TAU (*n* = 70)	0.64	The SMS intervention was associated with significant improvement in medication adherence at the end of the 3-month intervention (*p* < 0.001).
Montes (2012) [59]	Morisky scale	Baseline, 3 and 6 months post intervention	SMS intervention (*n* = 100) TAU (*n* = 154)	*n*/A	A significantly greater improvement in adherence was observed among participants receiving SMS text messages compared with the control group on the basis of self-reports after 3-month (*p* = 0.02) and after 6-month follow-up (*p* = 0.04).
Velligan (2013) [63]	-MEMS ^2^ -Pill counts	9 months	Med-eMonitor (*n* = 48) PharmCAT (*n* = 47) TAU (*n* = 47)	0.98 1.03	The two different behavioural interventions showed a statistically significant enhancement in medication adherence at all time points through treatment and after 9-month follow-up when compared with the control group (*p* < 0.001). Differences between the two behavioural interventions were not significant (*p* > 0.43).

^2^ Medication Event Monitoring System.

**Table 4 ijerph-18-10213-t004:** Summary of results on the effectiveness of educational interventions.

Reference	Assessment	Follow-Up	Number of Participants	Cohen’s d	Study Results
Aho-Mustonen (2010) [52]	Compliance Rating Scale	Baseline and 3 months post treatment	Psychoeducation (*n* = 19) TAU (*n* = 20)	0.53	No significant difference in adherence was noted at the baseline (*p* = 0.81) and after 3-month follow-up (*p* = 0.86).
Awan Riaz (2017) [44]	Compliance Rating Scale	Baseline and 3 months	Intervention group (*n* = 53) TAU (*n* = 50)	*n*/A	At baseline, there were 24% participants in intervention while 46% in control group who had complete adherence rate (*p* = 0.022). At 3-month follow-up, there were 96% cases in the intervention group and 47% in the control group with complete adherence (*p* < 0.001).
Bäuml (2016) [54]	-A four-step ordinal scale -Plasma drug levels	24 months and 84 months	Intervention group (*n* = 21) TAU (*n* = 20)	*n*/A	There were no significant differences found in adherence between the groups (*p* = 0.09).
Barkhof (2013) [53]	Medication Adherence Questionnaire	Baseline, 6- and 12 months	Health education (*n* = 59) Motivational interviewing (*n* = 55)	0.0	No significant differences were found between motivational interviewing and health education on the two adherence measures (*p* = 0.34).
Çetin (2018) [55]	-Medication Adherence Rating Scale -Morisky scale	Not reported	Intervention group (*n* = 55) TAU (*n* = 80)	0.56	The mindfulness-based intervention was associated with significant improvement in medication adherence (*p* < 0.05).
Eker (2012) [56]	Medication Adherence Rating Scale	2.5 months	Psychoeducation group (*n* = 35) TAU (*n* = 36)	*n*/A	The participants’ adherence in the psychoeducation group significantly increased (86.7%) after psychoeducation (*p* < 0.01).
Javadpour (2013) [48]	Medication Adherence Rating Scale	Baseline, 6, 8, 12 months	Psychoeducation group (*n* = 54) TAU (*n* = 54)	*n*/A	A statistically significant enhancement in medication adherence was found in the intervention group compared to the control group (*p* = 0.008).
Kopelowicz (2012) [43]	Unclear assessment tool	Baseline, 4, 8, 12, 18 and 24 months	Education (*n* = 64) Mixed (*n* = 53) TAU (*n* = 57)	*n*/A	The education intervention showed a statistically significant higher medication adherence than the mixed group after 18-month follow-up (*p* = 0.01) but not at 24 months (*p* = 0.20). More participants in education group were fully adherent than those in TAU at all assessments (*p* < 0.01).
Moncrieff (2016) [58]	Medication Adherence Questionnaire	1 and 3 months post intervention	Intervention group (*n* = 27) TAU (*n* = 23)	*n*/A	Participants in the education group indicating a greater tendency to be adherent with medication compared to those in the control group.
Sajatovic (2018) [64]	-TRQ -MEMS	Baseline, 10 weeks, 14 weeks and 6 months	Education group (*n* = 92) Mixed group (*n* = 92)	0.88	The education intervention showed a statistically significant lower medication adherence than the mixed intervention group (*p* = 0.048).
Valenstein (2009) [62]	-Medication Possession Ratios (MPR) -Composite Adherence Measure (CAM)	6- and 12 months	Education group (*n* = 58) TAU (*n* = 60)	0.76	Patients in the education group indicated a statistically significant higher adherence at 6- and 12-month follow-up compared to TAU (*p* < 0.001).

**Table 5 ijerph-18-10213-t005:** Summary of results on the effectiveness of mixed interventions.

Reference	Assessment	Follow-Up	Number of Participants	Cohen’s d	Study Results
Dahan (2016) [46]	Visual Analog Scale for Assessing Treatment Compliance	Unclear	Intervention group (*n* = 31) TAU (*n* = 32)	0.75	No significant differences between intervention group and TAU in medication adherence (*p* > 0.05).
Guo (2010) [47]	Unclear	12 months	Intervention group (*n* = 633) TAU (*n* = 635)	*n*/A	Non-adherence was noted in 2.8% of participants in the mixed intervention group and 5.7% in the control group (*p* = 0.006).
Kopelowicz (2012) [43]	Unclear assessment tool	Baseline, 4, 8, 12, 18 and 24 months	Mixed group (*n* = 53) Education group (*n* = 64) TAU (*n* = 57)	*n*/A	The mixed intervention showed a statistically significant lower medication adherence than the education group after 18-month follow-up (*p* = 0.01) but not at 24 months (*p* = 0.20). There was no significant difference at any point between the mixed intervention group and the TAU.
Pakpour (2017) [50]	-Medication Adherence Rating Scale -Plasma levels	Baseline and 6 months post intervention	Intervention group (*n* = 134) TAU (*n* = 136)	0.84	Measured by the Medication Adherence Rating Scale, the intervention group showed a significantly higher medication adherence compared to TAU both 1 month (*p* < 0.001) and 6 months (*p* < 0.001) after the intervention. Analysis of the plasma levels indicated that participants in the control group had slightly decreased levels at 6 months post intervention, suggesting that they may not have been adhering to their medication regimen. After controlling for study centre and repeated measurement, participants in the intervention group had significantly higher plasma levels of mood stabilisers than did participants in the control group at 1-month (*p* < 0.001) and 6-month (*p* < 0.001) follow-up post intervention.
Sajatovic (2018) [64]	-TRQ -MEMS	Baseline, 10 weeks, 14 weeks and 6 months	Mixed group (*n* = 92) Education group (*n* = 92)	0.91	The mixed intervention showed a statistically significant higher medication adherence than the education intervention group after 6-month follow-up (*p* = 0.048).
Schirmer (2015) [60]	-Pill count -Serum levels -Self-reported of medication intake at (unclear assessment tool)	1 month	Intervention group (*n* = 52) TAU (*n* = 50)	*n*/A	The intervention group showed a statistically significant higher medication adherence than the control group: 98% of the participants in the intervention group versus 76% in the control group were rated as adherent by pill count (*p* < 0.01). By measurement of serum levels, 88.5% versus 70% were adherent (*p* < 0.05).

**Table 6 ijerph-18-10213-t006:** Risk of bias assessment and quality assessment.

Outcomes	Anticipated Absolute Effects * (95%CI)		Relative Effect (95% CI)	No of Participants (Studies)	Quality of Evidence (GRADE)
Effects on adherence (behavioural interventions) assessed with: MAQ, MARS, MEDAD, MEMS, Morisky and pill counts. Follow-up: range 1,5 month to 15 months.	Risk with no intervention or other intervention	Risk with adherence-enhancing intervention	-	1059 (9 RCTs)	Very Low ^1,2,3^
Effects on adherence (educational interventions) assessed with: CRS, MARS, MAQ, Morisky, MPR, MEMS and TRQ. Follow-up: range 1 month to 7 years.	No estimable see comments	No estimable see comments	-	1134 (11 RCTs)	Very Low ^1,2,3^
Effects on adherence (Mixed interventions) assessed with: MARS, MEMS, plasma concentrations, pill counts, TRQ and VASTEC. Follow-up: range 1 month to 24 months.	No estimable see comments	No estimable see comments	-	2045 (6 RCTs)	Low ^1,2^

* The risk in the intervention group (and its 95%confidence interval) is on the basis of the assumed risk in the comparison group and the relative effect of the intervention (and its 95%CI). CI: Confidence interval; RR: Risk ratio; OR: Odds ratio. GRADE Working Group grades of evidence. High quality: We are very confident that the true effect lies close to that of the estimate of the effect. Moderate quality: We are moderately confident in the effect estimate: The true effect is likely to be close to the estimate of the effect, but there is a possibility that it is substantially different. **Low quality**: Our confidence in the effect estimate is limited: The true effect may be substantially different from the estimate of the effect. **Very low quality**: We have very little confidence in the effect estimate: The true effect is likely to be substantially different from the estimate of effect. ^1^ Downgraded due to unclear risk of bias for allocation concealment, blinding of participants and outcome assessors or both. ^2^ The quality of the evidence of the studies measuring this outcome was downgraded due to the lack of precision or lack of consistency, or both. ^3^ Downgraded due to high risk of bias for allocation concealment, blinding of participants and outcome assessors or both.

## Data Availability

All data is contained within the article and Appendix A.

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
