# Peer review of "Interventions to Improve Medication Adherence in Patients with Schizophrenia or Bipolar Disorders: A Systematic Review and Meta-Analysis"

_ijerph, 2021, doi:10.3390/ijerph181910213_

Round 1

Reviewer 1 Report

This manuscript explores an important topic, but was a bit difficult to follow and confusing in some areas. I have a few specific comments for your consideration to ensure clarity to the reader:

  • The Abstract does not mention findings of meta-analysis.
  • The conclusion in the Abstract is very brief.
  • There are minor issues with grammar and sentence structure throughout. I think it needs to be reviewed and edited for English language.
  • I think the aim needs to be reworded. It currently states that the effectiveness of interventions that “improve” adherence was explored. However, I believe it should state that the impact of interventions on medication adherence was explored (i.e. some of the included interventions may not have improved adherence, even if they aimed to). I think the first sentence under “electronic searches better describes the “aim”.
  • Please explain the difference between assessing the quality of evidence and the certainty of evidence, and make this clear in the Methods and Results. In the methods, I think the description of using GRADE should be under its own subheading (i.e. it is not an assessment of “Risk of bias” and should not be under that subheading)
  • Why was the date range 2009-2019 chosen? Was another review identified that had similar aims and explored evidence until 2009? 2019 is now two years ago, was an update considered to identify evidence published since?
  • Why was the search in Pubmed limited by study type but not in Web of Knowledge? By applying this filter you may have missed out on longitudinal and pre/post studies – were these excluded?
  • Please clarify whether people with self-reported diagnoses were considered eligible for inclusion, or did the study have to specify that a diagnosis was made a by a doctor as per the DSM or ICD-10. And please specify which edition of the DSM was used.
  • It is unclear if only studies with a primary outcome measure of adherence were included or if any studies exploring the impact of an intervention on adherence (i.e. even as a secondary outcome) were included. The Results indicate that 23 studies were excluded as they did not report on adherence – did they not report on adherence at all? Or did they not have adherence as a primary outcome measure?
  • What is meant by “segmented publications”?
  • Page 4, Line 147: Do you mean that 23 studies were “included”?
  • Please use exact statistics (i.e. what is meant by “about 70%”?)
  • It is unclear if “family involvement, intensified patient care” falls under “behavioural” or “education” or “mixed” in Table 3.
  • Lines 196-200: Please re-word as this sentence does not have a verb.
  • The paragraph about “Selective Reporting” mentions “seizure frequency” – is this meant to be an example as it is a bit confusing as to why “seizures” are mentioned specifically?
  • The title of Table 4 is not descriptive enough. Is it meant to demonstrate a combination of the risk of bias assessment and the quality assessment? It seems these have been combined in-text as well, which is a bit confusing to the reader. I suggest separating to improve clarity and flow.
  • I suggest reordering the results section and moving the risk of bias and quality assessment sections to the end.
  • This sentence “All interventions aimed at improving medication adherence, however, the intervention was unclear (21), one study focused on general health (53) and one on diagnosis and identification of recovery-informed therapy goals (57)” requires further explanation. Was the intervention “unclear” in one study (21) or across all studies? The focus on “general health”, “diagnosis” and “goals” – was this in addition to the primary outcome of “adherence”?
  • Lines 316-317: Please re-word this sentence as it is unclear and provide references.
  • Lines 319-320 are a repetition of lines 323-325.
  • Line 327 is a repetition of Line 322.
  • Lines 332-333: Please re-word this sentence as it is unclear and provide references.
  • When interventions are described as having a “positive” impact, it is unclear if this was statistically significant.
  • Abbreviations need to be spelt out when first mentioned in-text (e.g. SMS, Q&A).
  • Lines 360-362: Please re-word this sentence as it is unclear and provide references.
  • In the first paragraph of the sub-heading “Effects on adherence”, you mention that effect sizes were available/could be calculated for 15 interventions but could not be for 14 interventions. This gives a total of 29 “interventions” but there are only 23 studies included in the review. Then in the discussion, it states that there were 30 interventions. Please clarify and please try to use consistent terminology as in some places it seems the term “intervention” and “study” are being used interchangeably.

Discussion:

  • I think it is important to recognise that there are other systematic reviews/meta-analyses published that have had similar aims to this one and to explain how the current systematic review is distinct. For example:
    • MacDonald, L., Chapman, S., Syrett, M., Bowskill, R., & Horne, R. (2016). Improving medication adherence in bipolar disorder: A systematic review and meta-analysis of 30 years of intervention trials. J Affect Disord, 194, 202-221. doi:https://doi.org/10.1016/j.jad.2016.01.002
    • Yaegashi, H., Kirino, S., Remington, G. et al.Adherence to Oral Antipsychotics Measured by Electronic Adherence Monitoring in Schizophrenia: A Systematic Review and Meta-analysis. CNS Drugs 34, 579–598 (2020). https://doi.org/10.1007/s40263-020-00713-9
    • García, Saínza et al. “Adherence to Antipsychotic Medication in Bipolar Disorder and Schizophrenic Patients: A Systematic Review.” Journal of clinical psychopharmacology4 (2016): 355–371. Web.
  • This Discussion is largely just a summary of the findings and detailed descriptions of some of the included studies. The Discussion would benefit from explanation of the findings of the current review in the context of the broader literature.
  • If authors were contacted to clarify missing information, this should be described in further detail in the methods and results.
  • Lines 265-269 and 462-465 are the same.
  • There are instances where the term “schizophrenic” is used, and this should be replaced with “schizophrenia” or “people living with schizophrenia” where appropriate.
  • Please review reference list (e.g. “Organisation WH” should be “World Health Organisation”).

Reviewer 2 Report

Overall, this is a good area to review and I commend the authors for their effort on this work. I have the following questions and recommendations for the authors to address.

Methods

  • On electronic searches -  It is unusual to use 'Web of Knowledge' if there is any database. Do you mean Web of Science?
  • Is there any reason why you restricted your search timeframe to 2009-2019?
  • It is unusual to have a table in the method section. I recommend you take this to appendix.
  • Page 3 line 93 take out the word quantifications...
  • Again why is the data extraction limited to 2019? It's been 2 years since. There is a high chance you could be missing a study.
  • You mentioned the use of Cochrane Risk of bias tool, is this the latest version 2.0? Make this explicit along with the specific domains in the tool. Based on your findings this looks like the earlier version. Is there a reason why you didn't used the latest version? 

Results

  • On the flowchart, I didn't understand what the numbers in the bracket are in the studies included
  • Page 6, line 206 you mentioned "Most of the studies defined adherence as taking more than 70% of prescribed doses." How did you come up with this figure for the self-report measures ? These are mostly based on brief questionnaires with dichotomous responses. A good example if Morisky, which has 4-8 items. I don't know how you define 70 % in this case.
  • I think the findings of the RoB assessment is unnecessarily long. I recommend you shorten it to one to two paragraphs top. 
  • I found the tables in the results a bit difficult to follow. They need some restructuring and better description for better clarity. Table 3 needs major modification to make it easy to understand. Describe the components of the interventions and the contexts they were implemented in with greater details. Also, maybe add a separate column for the intervention types. But looking at Table 5-7, now I am thinking if you actually need table 3. It could be better to use textual narrative instead of table 3 as an alternative. 
  • You have table 1 then table 3. Where is table 2?
  • Have you considered looking at publication bias in your quantitative analysis to see the study variations?
  • generally I would be careful in drawing strong conclusions given the # of studies in the quantitative analyses are very limited.

Discussion

  • Page17 line 446 needs paraphrasing - I am not sure what you meant there.
  • You have this on page 17 line 455 "This methodology resulted in the inclusion of a large number of studies, contrary to previous reviews" ... please cite the reviews you are referring to here.

Reviewer 3 Report

This manuscript is about the systematic review and meta-analysis conducted to identify the effectiveness of interventions improving medication adherence in patients with schizophrenia or bipolar disorders. Overall, the study is conducted and written very well. Few specific comments on the study are as follows:

  1. In the abstract, authors are suggested to describe how the effectiveness of interventions was measured (mention that effect sizes-Cohen’s d was used).
  2. To describe the rationale of this study, the authors state that “To date, however, a comprehensive 50 assessment of the effectiveness of these interventions, improving medication adherence 51 in patients affected by schizophrenia or bipolar disorders, is lacking.” Conducting SLR will not assess if and how these interventions are effective to improve adherence, however, SLR and MA will help to identify different interventions used to improve adherence and to what extent they improve adherence, but results can not be used to assess the effectiveness of interventions. Authors are suggested to clarify this in their rationale and aim and add a statement in the limitation.
  3. Why retrospective studies were not included in the study?
  4. How conflicts between two independent reviewers were resolved?
  5. Line 340-341: The statement is not clear “All measured the effect on medication adherence.”
  6. In Table 7, under Cohen’s d, / is reported. What does that mean?
  7. Different medication adherence measurements and definitions were used by studies analyzed in this SLR/MA, hence the association between type of intervention and adherence cannot be established. Authors are strongly encouraged to discuss this major limitation.
  8. The study emphasizes that this is a comprehensive synthesis. Though this SLR is a thorough review, I don’t think it is a “comprehensive” review because the authors did not consider retrospective studies, did not cover studies published after November 2019 and did not cover studies published in all countries. Hence, authors are suggested not to claim that this study is “comprehensive”.

Round 2

Reviewer 2 Report

The reviewers did a good job addressing my questions. I am happy to recommend this for publication.

Author Response

Dear reviewer,

Please find attached the revised version of our manuscript entitled: ‘Interventions to improve medication adherence in patients with schizophrenia or bipolar disorders: A systematic review and meta-analysis’, submitted to International Journal of Environmental Research and Public Health. We would like to thank the reviewers.

We hope that we have sufficiently addressed your comments. Thank you for considering this manuscript for publicatie in the International Journal of Environmental Research and Public Health.

Sincerely yours,
On behalf of all co-authors,
Elke Loots